# Smart Organization of Imbalanced Traffic Datasets for Long-Term Traffic Forecasting

**DOI:** 10.3390/s25041225

**Published:** 2025-02-18

**Authors:** Mustafa M. Kara, H. Irem Turkmen, M. Amac Guvensan

**Affiliations:** Computer Engineering Department, Yildiz Technical University, Istanbul 34220, Türkiye; mmkara@yildiz.edu.tr (M.M.K.); irem@yildiz.edu.tr (H.I.T.)

**Keywords:** long-term traffic speed prediction, intelligent transportation systems, deep learning, data preprocessing, imbalanced datasets, data grouping, training enhancements

## Abstract

Predicting traffic speed is an important issue, especially in urban regions. Precise long-term forecasts would enable individuals to conserve time and financial resources while diminishing air pollution. Despite extensive research on this subject, to our knowledge, no publications investigate or tackle the issue of imbalanced datasets in traffic speed prediction. Traffic speed data are often biased toward high numbers because low traffic speeds are infrequent. The temporal aspect of traffic carries two important factors for low-speed value. The daily population movement, captured by the time of day, and the weather data, recorded by month, are both considered in this study. Hour-wise Pattern Organization and Month-wise Pattern Organization techniques were devised, which organize the speed data using these two factors as a metric with a view to providing a superior representation of data characteristics that are in the minority. In addition to these two methods, a Speed-wise Pattern Organization strategy is proposed, which arranges train and test samples by setting boundaries on speed while taking the volatile nature of traffic into consideration. We evaluated these strategies using four popular model types: long short-term memory (LSTM), gated recurrent unit networks (GRUs), bi-directional LSTM, and convolutional neural networks (CNNs). GRU had the best performance, achieving a MAPE (Mean Absolute Percentage Error) of 13.51%, whereas LSTM demonstrated the lowest performance, with a MAPE of 13.74%. We validated their robustness through our studies and observed improvements in model accuracy across all categories. While the average improvement was approximately 4%, our methodologies demonstrated superior performance in low-traffic speed scenarios, augmenting model prediction accuracy by 11.2%. The presented methodologies in this study are applied in the pre-processing steps, allowing their application with various models and additional pre-processing procedures to attain comparable performance improvements.

## 1. Introduction

Traffic speed prediction is a crucial and longstanding research domain within Intelligent Transportation Systems (ITS) [1]. Accurately predicting traffic speed has the potential to mitigate numerous severe issues, including extended travel times, a high incidence of accidents, and carbon emissions. Additionally, it benefits city planning and countless other ITS applications [2]. The issue of speed prediction encompasses two fundamental characteristics. One of these characteristics pertains to the road network and the inter-dependencies among various route segments. A segment may precede or follow another segment with distinct characteristics or be entirely unconnected geographically yet exhibit nearly comparable patterns. Another element is temporality, which includes not just the periodicity of traffic influenced by the day of the week but also factors such as weather, special occasions, or recurring occurrences like the start of the school semesters. Initial research into this topic mostly concentrated on the temporal attributes of traffic, particularly the historical speed component, as they can be easily learned using statistics and basic machine learning models [3]. Subsequently, with the progress in deep learning methodologies, increased integration of temporal and spatial features was employed, mostly together, to forecast traffic speed [4].

Another aspect of traffic speed prediction that has evolved is the prediction horizon. The majority of research on this topic concentrates on short-term forecasts that operate within an acceptable margin of error, generally ranging from 15 to 60 min and extending to several hours [5]. Over the years, we utilized not only thousands of hours of speed data for training but also integrated additional data such as weather [6,7], accidents [8], holidays and special events [9,10], and road topography [11]. These changes enabled predictions for days [12] or even weeks [13] ahead of time. Despite the incorporation of extra data sources, forecasting traffic speed one week in advance continues to be a challenging issue, with existing literature predominantly focused on short-term prediction research [14]. Several studies addressing this issue [12] typically possess a prediction horizon of one day and exclusively employ short-term forecasting methodologies, which may be inadequate for long-term predictions. A further complication with long-term speed forecasting is necessary for more historical data. In contrast to short-term prediction, which predominantly depends on immediate preceding data, known as short-term dependencies, long-term prediction is based on extensive past data and necessitates a greater volume of past data for accurate forecasting. This requirement also affects the resemblance of historical data used in the input to the predicted label because as the window of history lengthens, the similarities between previous and future data decrease.

A prevalent issue seen in both long-term and short-term forecasting is the presence of imbalanced datasets. Traffic speeds vary during the day, and although the nighttime traffic flow rate is significantly high, representing one-third of the day, the periods of congestion, characterized by reduced speed, are confined to a few hours. This disparity is present in every traffic dataset and impacts prediction accuracy. Our tests revealed that models designed for long-term predictions with a one-week horizon were adversely affected by this phenomenon. Predictions from models trained on four months’ worth of imbalanced historical speed data are typically greater than what should be expected at low traffic speeds.

Oversampling and undersampling techniques are prevalent in the literature for addressing the issue of imbalanced datasets. Undersampling approaches may lose effectiveness because they exclude crucial samples from the majority group, causing a bias against it [15]. Conversely, oversampling presents certain drawbacks. Oversampling results in an enlarged training set due to the duplication of patterns, which consequently extends learning durations and elevates the likelihood of over-fitting [16]. However, in our experiments, we discovered that these conventional methodologies are not effective when dealing with traffic speed data. Therefore, domain-specific strategies are utilized not only to alleviate dataset imbalance but also to acquire training data that accurately reflects the patterns of the samples to be forecasted.

This work presents three innovative ways to address the limitations of an imbalanced traffic dataset, hence enhancing the efficacy of long-term traffic speed prediction.


**Hour-wise Pattern Organization**
The hours of a day exhibit constant patterns regardless of external circumstances. Hour-wise Pattern Organization entails the categorization of patterns by dividing the day into distinct segments based on hours that exhibit analogous patterns, followed by the training of a model for each segment.
**Speed-wise Pattern Organization**
Training models at similar speeds enhances the model’s ability to comprehend the relationships within the data. We conducted pattern organization by segmenting speed data with boundaries that encapsulate analogous patterns and trained a model for each of these clusters.
**Month-wise Pattern Organization**
Training models using immediate preceding months produces models that learn patterns that do not match the tested month (for example, using patterns from August to learn September). Month-wise Pattern Organization entails the structuring of patterns utilizing similar months for model training. This method can additionally be employed alongside both Speed-wise Patterns Organization and Hour-wise Pattern Organization to further enhance accuracy.

This work is, to our knowledge, the first research employing this type of organization to address imbalanced traffic dataset issues and enhance long-term traffic speed forecasting. The approaches presented in this research utilize traffic speed data exclusively to illustrate their adaptability for application with any model or dataset that may include supplementary variables. The proposed strategies are straightforward to adopt during the preprocessing step and do not negatively impact the training process, enhancing the accuracy at no cost. We demonstrate the impact of the proposed methods in our experimental results by using four models with established success in traffic speed prediction: LSTM, Bi-Directional LSTM, GRU, and CNN. This paper concentrates on preprocessing techniques aimed at enhancing long-term predictive performance. We have selected to experiment with the previously mentioned fundamental models. Firstly, to illustrate the efficacy of our methods across various models, and secondly, to validate our methods without the confounding influence of additional variables that complex models may introduce.

In the next section, we go over the research done in traffic speed prediction and review certain changes that happened in this domain. In Section 3, we introduce the dataset used in this paper, while in Section 4 we explain the proposed techniques in detail and show the results by using a simple LSTM network for long-term speed prediction utilizing the traffic data collected from a metropolitan city, Istanbul. In Section 5, we test our proposed methods with four different deep-learning architectures to show their effectiveness with different model types. In Section 6 we discuss our research and its short comings. And in the Section 7, we conclude our work and give a discussion about how this research might evolve in the future.

## 2. Literature Review

Traffic forecasting is an important subject in intelligent transportation systems. As it has become a popular topic, many studies with statistical methods were published at an early stage. Among these models, ARIMA (auto-regressive integrated moving average) has achieved the most success and remains a benchmark due to its ability to capture the periodicity of traffic data and its robustness to outliers. Ahmed et al. [17] found that ARIMA worked very well when they compared it to other statistical models like moving average, double-exponential smoothing, and Trigg and Leach adaptive models. Later on, variations of ARIMA also became popular in literature. Van Der Voort et al. [18] used ARIMA with Kohonen maps to divide days into different clusters and build an ARIMA model for each cluster. Although ARIMA has remained a hot topic among researchers, most of the attention has shifted to more complex models with the rise of machine learning algorithms. Researchers have extensively explored k-NN (k-nearest neighbor) and SVR (support vector regression) models for predicting traffic speed and flow [19,20,21,22,23].

At some point, researchers started to think of traffic prediction as a non-linear problem, and with the availability of data increasing over the years, neural networks and deep learning methods became the focus of the traffic prediction problem. Refs. [24,25,26] are early works exploiting neural networks for short-term traffic prediction. More recent papers [27], have benefited from neural networks with ensemble model structures along with the average speed of the previous weeks.

Although neural network models could yield good results, they could hardly capture the complexity of traffic data, which has both spatial and temporal qualities. To capture such features, CNN and LSTM variants [28] are widely utilized in current research studies. Refs. [29,30,31] are early papers that used LSTMs. They didn’t make many changes to the original LSTM model or add any new steps to prepare the data for LSTM to use it on traffic data. Due to the compatibility of LSTM and traffic data, even these types of models could produce better results than statistical or machine learning models. However, in recent years, researchers have employed various techniques to enhance prediction accuracy. GRU (Gated Recurrent Units) [32] is a recurrent neural network model that works similarly to LSTM’s but has a lower amount of variables to learn data relations. Fu et al. [33] compared LSTM and GRU, and they found that GRU performed better than LSTM on traffic flow prediction. Another interesting change to the LSTM structure was the attention mechanism [4,11,34,35,36,37,38,39], which became a de facto technique for LSTMs and GRUs.

CNN, on the other hand, is better at capturing spatial properties in traffic data. Different road segments and their relations with each other significantly contribute to expediting prediction. However, defining the relationships between roads is necessary for this process. Earlier work using CNNs took this relation as a grid map to build this relation with the help of Euclidean. In a different way, Zang et al. [40] use convolution layers in the Generative Adversarial Network(GAN) structure to predict daily speed through the generator. They then train the generator with a discriminator that compares real speed data with the generated data. To test this model, the authors used one year of data from two road segments and made predictions for the day after. They achieved a 32 MAE (Mean Absolute Error) in high-traffic, low-speed moments. Mendez et al. [41] used CNN and Bi-LSTM together by using CNN to extract hidden temporal relations while making Bi-LSTM learn these relations. They were able to achieve an 8% improvement in prediction performance over a 3-day prediction horizon. Ma et al. [42] uses a similar logic, but rather than using CNN for extracting spatial relations, a method called STFSA (spatial-temporal feature selection algorithm) is used instead. The CNN-GRU block then received the results of the STFSA as input.

Regular convolutional methods fail to grasp the nature of spatial information in traffic data. Since roads in a city build a graph, a grid cannot capture this complex relationship. Thus, Graph CNN (GCN) [43,44] were used extensively in traffic prediction. In [12], graph convolution layers were exploited to guess traffic flow a day later with a MAPE (Mean Absolute Percentage Error) score of 10. Most recent models leverage temporality in conjunction with GCN to effectively capture these two characteristics. Bogaerts et al. [45] find similarities to roads before training; it first uses a GCN to learn these spatial relationships and then uses LSTM to capture temporal information. Han et al. [46] use daily traffic data to find other similar roads and apply this similarity to quantify the effect of each road on the predicted part of the day. Chen et al. [47] used visual data to quantify traffic flow and then used GCN and GRU to extract spatial and temporal features. In their work, they utilized downflow information to predict sudden changes in the traffic and managed to improve prediction accuracy on 30-min predictions by 17% compared to state-of-the-art models. Qi et al. [48] used environmental variables along with temporal variables and then used a spatio-temporal graph convolutional network for predicting traffic speed 3 h later. They managed to outperform other spatio-temporal graph models by 5% in their experiments.

Relationships (similarities) between the roads are hard to find with statistical methods. Kara et al. [49] used a variation of the k-nearest neighbor algorithm to find similar road segments in order to fill missing values. State-of-the-art models, on the other hand, utilize dynamic correlation between roads and learn this relationship while training a prediction model. In [11], researchers used a dataset from 2015 that has speed values for 5-min windows. They used a model that is a mix of GRU and Graph Convolution Layers, and it learns from its own attention how the k-hop neighbors affect the current road segment. It predicts 60 min ahead with a MAPE value of 2. In another study, an attention mechanism is employed to identify road relations [38]. However, unlike other papers, it does not define a connected graph but learns the relations of every road on the map. Wang et al. [50] use a similar idea but with the addition of a multi-scale attention module to also extract the temporal relations and then combine it with the dynamic spatial relations. Qui et al. [51] developed a graph model that takes events into account for traffic speed prediction that achieved an 8.09% MAPE score for a two-hour prediction.

We have so far discussed the history of speed prediction techniques. However, our analysis of traffic speed data revealed that the distribution of speed values leads to a significant issue known as the data imbalance problem, which poses a significant challenge for the deep learning domain. In the real world, it is common to encounter such imbalances across various datasets, specially collected by sensors. There are two methods frequently employed to deal with this problem. Both of these methods involve modifying the dataset. Undersampling is a method that selects samples to ensure an equal or balanced distribution of classes [16], or continuous labels in regression problems. The other method is called oversampling, where new samples are produced using existing samples through different methods [52].

A widely utilized oversampling technique is the Synthetic Minority Oversampling Technique (SMOTE) [53] for imbalanced classification issues and the Synthetic Minority Over-Sampling Technique for Regression (SMOTER) [54] for imbalanced regression problems. SMOTE employs statistical methods to generate more samples for the minority class. Throughout the years, various iterations of these techniques have been published in the literature, each with distinct focuses, such as Gavas et al. [55], who incorporated spatial information to balance the dataset, and Camocho et al. [56], who developed weights for each sample based on their rarity to use in SMOTER calculations. Despite the widespread application of these oversampling strategies, their efficacy is not universally applicable. Avelino et al. conducted a survey comparing resampling approaches, including SMOTER and its modifications, over 30 distinct imbalanced datasets. Two conclusions may be drawn from the findings of this paper: first, in numerous instances, the absence of a balancing strategy yielded the highest accuracy; second, the optimal resampling technique varied depending on the dataset, indicating that no universal strategy exists. These conclusions indicate that data balance is significantly influenced by the dataset and the context of the prediction problem.

The advancement of data creation techniques, such as variational autoencoders (VAEs) and GAN models, has shifted the emphasis regarding oversampling and data augmentation strategies toward deep learning methodologies. Abdulganiyu et al. utilized a combination of class-wise focal loss and variational autoencoder for data augmentation on network traffic data. Engelmann et al. [57] employed Conditional Wasserstein GAN for oversampling, whereas Fan et al. [58] utilized CycleGAN for data augmentation. Sharma et al. utilized a hybrid approach that integrates GAN and SMOTE methodologies. The authors first use SMOTE to generate synthetic samples and then employ a GAN model to enhance the quality of these artificial samples. This approach effectively integrates both techniques to overcome their respective limitations. While data generation methods effectively balance datasets, they are not well suited for the domain of traffic speed prediction. Traffic speed datasets are too small to train data-generating models such as GANs. The lack of data lowers the quality of synthetic data, and the low number of minority classes in traffic speed datasets makes it even harder for models to find underlying patterns in traffic, which results in synthetic data that are not realistic.

Alternative approaches for addressing unbalanced datasets employ various strategies that do not involve balancing the dataset, which include specific models or training processes that mitigate bias. Guo et al. [59] used a focal loss-based approach for adaptive gradient boosting. Ren et al. [60] employed a modified Mean Squared Error (MSE) to address the imbalanced dataset, whereas Steininger et al. [61] developed weights for samples based on their rarity; however, instead of applying these weights in a technique such as SMOTER, the authors utilized them to adjust the loss value for each sample.

Domain-specific balancing methods are necessary in all domains but especially traffic speed prediction. Furthermore, the distinction in methodology that is required when it comes to long- and short-term predictions extends to this problem. Since the same reasons that limit the effectiveness of short-term prediction methods for long-term forecasting also apply, similar concerns should be addressed in the context of data balancing strategies. However, the current traffic speed prediction literature primarily focuses on models that extract the best spatio-temporal features rather than addressing the inherent problems in the traffic speed datasets. Few papers that specifically focus on this problem either try to tackle it for accidents and anomalies in the dataset [62] or focus on short-term predictions [63,64]. To the best of our knowledge, we are the first research that focuses on the data imbalance problem for long-term traffic speed predictions. Our proposed methodology does not involve removing or creating samples but focuses on reorganizing the training dataset to extract useful patterns and increase the learning capacity of the prediction model. We propose three different ways to reorganize the dataset: hour-wise, to emphasize hourly characteristics; speed-wise, to emphasize the temporal relations of speed values; and month-wise, to emphasize seasonal changes brought about by weather. Our methods serve as a preprocessing step, allowing for compatibility with various model types and other preprocessing steps. This flexibility and robustness enable them to seamlessly integrate with various methodologies and features without incurring additional computation costs.

## 3. Dataset

All experiments were conducted using a traffic speed dataset gathered by the Istanbul Municipality in 2018. Table 1 presents the specifications of the dataset. This dataset contains traffic speed data, dates, and location information for 441 primary road segments in Istanbul, all segments averaging 1 km in length. A total of 111 segments were chosen from 441 segments to reduce the training time. The selection involved choosing one from four successive segments, enabling an accurate and equitable representation.

All data were gathered at one-minute intervals. Each road segment comprises 525,600 traffic speed data points. Certain data were absent due to sensor errors. To address the missing data and enhance the accuracy of our prediction problem [65], we modified the data interval to 5-minute segments by averaging 5-minute windows. Following this step, we employed a moving average window of size 6, containing data from the preceding 30 min, to impute the remaining missing values. Given the minimal missing values, we eliminated them from the training and testing datasets. Given that traffic patterns are consistent on the same weekdays, our dataset comprises the last 6 weeks of data corresponding to the specific day of the week. All models in the methodology and the experiment results were trained to forecast up to one week ahead utilizing historical data from the preceding six weeks, including 15 min before and after the predicted time. It is important to note that the train data have traffic speed values weeks before the label data; therefore, using 15 min after the expected time of day does not result in a data leak.

## 4. Methodology

Traffic speed prediction is based on the periodic characteristics of traffic, and models designed for this problem attempt to capture this by utilizing previous speed data or supplementary variables such as the date. The more the model learns these features, the better the forecast results will be. To improve results, training models for common patterns is a feasible choice. Previous research on traffic prediction lacks data to measure such similar patterns. The current practice in the literature ([11,38], etc.) is to find similarities between road segments to achieve successful model training. This study presents three distinct ways of representing the general characteristics of traffic data. The fundamental concept underlying these three strategies is to identify similar historical data and categorize them to highlight traffic trends. These organizational schemes are predicated on temporal factors, velocity, and monthly data. To demonstrate their efficacy, the proposed methodologies are compared to a base model with no pattern -rganizing procedure applied to it.

Figure 1 presents a general overview of the proposed methodology. The process starts with an imbalanced traffic speed dataset. This dataset is then organized according to one of the three proposed methods. It is worth noting that these strategies can be combined, as stated in the introduction. Following the determination of the methodology to be employed, a more balanced dataset is produced, which has been divided into different groups, each of which exhibits similar characteristics. After pattern organization, model training occurs on the balanced dataset. In this paper, we experimented with four different deep learning models: LSTM, CNN, bi-directional LSTM, and GRU, although our methods are applicable to other machine learning and deep learning models as well. After model training, generated models were collected into a pool. The test data are processed using the chosen pattern organization technique to balance the dataset and are then assigned to the most suitable model, trained on data with characteristics similar to the test sample. Following the model selection, we obtain the predicted traffic speed value.

The proposed methods categorize the dataset according to three strategies: Hour-wise, Speed-wise and Month-wise Pattern Organization. Each strategy relies on separating the dataset into tailor-made groups that have high in-class consistency and the capacity to adequately represent the general characteristics of the group. In the training phase, a separate prediction model is created for each group. In the testing phase, samples are assigned to the model based on the features utilized to form distinct groups. The Hour-wise Pattern Organization utilizes the time-of-day feature to form five distinct groups. This approach organizes the dataset into discrete categories according to the hour of sampling. Samples are segregated identically during both the training and testing phases. In the context of speed-based pattern organization, the data were divided into four distinct groups. During the training phase, the label values were utilized to establish separate groups; however, in the testing phase, where direct label usage is not permitted, the average of the input speed values was employed to select the most appropriate prediction model. We also establish overlapping groups in the training phase to ensure that distinct models created for each speed group also incorporate samples from other speed groups, should there be a need to forecast such volatile samples. Month-wise Pattern Organization identifies the most similar months for training, ensuring that the training and test datasets exhibit similar characteristics. Our suggested methodology achieves this by creating a representation for each month and making comparisons among them. After identifying similar months, we trained a model for each month using the most similar months.

Both Mean Absolute Percentage Error (MAPE) and Mean Absolute Error (MAE) are widely utilized metrics to illustrate the performance results of forecasting algorithms. We noticed that MAPE values provide a more accurate representation of model performance, as they emphasize errors occurring during peak traffic periods. Consequently, we chose MAPE to demonstrate the efficacy of our approach.(1)$$\mathrm{MAE}\left(y,\hat{y}\right)=\sum_{i=0}^{N-1}\left|y_{i}-\hat{y_{i}}\right|$$(2)$$\mathrm{MAPE}\left(y,\hat{y}\right)=\frac{1}{N}\sum_{i=0}^{N-1}\frac{y_{i}-{\hat{y}}_{i}}{y_{i}}.$$(3)$$\mathrm{Loss}\;\mathrm{Function}=\frac{3*\mathrm{MAPE}+\mathrm{MAE}}{4}$$

During the training process, we employed a combination of MAE and MAPE metrics. We have tested each separately and noted that using them individually results in each of these error rates introducing bias into the model. The formulas for these two measures are presented in Equations (Equation 1) and (Equation 2). These equations provide insight into the cause of the bias. MAE values represent the absolute disparity between the actual and projected sequences. The MAPE metric, on the other hand, computes an error rate that is proportional to the real value. Consequently, MAPE has an elevated error rate when the real values are lower. When utilized as a loss function, this contrast causes MAE-trained models to output higher values and MAPE-trained models to output lower values. In light of this bias, we have opted to build a custom loss function that integrates both measures. The proposed custom loss function is presented in Equation (Equation 3). We utilize several coefficients in our calculations to assign greater significance to the MAPE score. We have noted that the implementation of this custom loss function achieved less bias in the trained models.

The results presented in this section are obtained from LSTM-based models. These models have a single LSTM layer, followed by a dense layer and an output layer. The LSTM layer has 256 hidden units, whereas the dense layer has 128. Each layer incorporates a dropout layer with a dropout rate of 0.4. The dense layer undergoes batch normalization, whereas the LSTM layer does not.

### 4.1. Training Models with Hour-Wise Pattern Organization

The most clear trend that traffic data shows is related to the time of day. The most prominent illustration of this trend is the rush hours, which lead to significant traffic congestion irrespective of the day of the week. Another case is nighttime, when no traffic congestion can be seen. While weather and special occasions create variations, such diversions primarily influence the magnitude of speed rather than its exhibited pattern.

Figure 2 illustrates that specific periods in a day exhibit similar characteristics. Consequently, we decided to partition 24 h into five distinct intervals. The initial period spans from 05:40 to 09:40, the subsequent period extends from 09:40 to 14:00, the third period lasts from 14:00 to 19:00, the fourth period ranges from 19:00 to 22:00, and the final period occurs from 22:00 to 05:40. We created these subsets to separate increasing and decreasing characteristics so that models could learn them more effectively. Figure 3 presents the percentage of data corresponding to each of these temporal subsets. For each of the five periods, a distinct model was used, and to forecast a specific hour of the day, the model corresponding to the hour in question was applied.

An illustration of this procedure is depicted in Figure 4. This graphic displays a matrix including six rows of data from the past week and seven columns representing 30 min intervals of traffic speed data. Furthermore, the labels of these samples and their corresponding dates are also included in the same figure. Based on the temporal data of the associated sample, we determine the time group and which model these samples will train and test. In this instance, samples were collected at approximately 12 p.m., categorizing them under the Noon Model, which includes the hours from 10 a.m. to 2 p.m.

To evaluate the performance of this method, a model is trained on the entire dataset, and the outcomes are compared to those of a model trained using the proposed technique. Table 2 presents the outcomes of the base LSTM model and the results of the suggested pattern-organizing techniques. Hour-wise Pattern Organization improved normal training MAPE by only 0.2%. Figure 5 presents the data obtained on an hourly basis. The upper graphic illustrates a comparison between the normal training process using an LSTM model and the proposed Hour-wise Pattern Organization training approach. The lower graphic shows the disparity between these two training methodologies, with green areas denoting the intervals during which Hour-wise Pattern Organization has surpassed the normal training procedure, and red areas showing where it has underperformed. This figure reveals that, despite a low average improvement, there is a significant improvement of 0.8 percent MAPE during peak rush hours.

### 4.2. Training Models with
Speed-Wise Pattern Organization

Traffic speed datasets have an intrinsic imbalance, with low-speed samples comprising merely 11% of the overall data. Figure 6 illustrates the speed distribution of our dataset. The spike at 60 km/h is because of the default value assigned in the speed collection process. Figure 7 illustrates the percentages of low-, normal-, and high-density traffic states. This imbalance complicates learning from these datasets, and most models predict greater values than usual speeds. To address this issue, we trained various models according to distinct speed categories. The primary idea behind this strategy is to divide the dataset into speed groups and then train a separate model for each group. The key aspect of this process involves determining the speed limits. Two significant issues must be considered. The initial issue is the volume of data inside each group. Because deep learning methods require a particular quantity of data, if a group has insufficient training samples, the outcomes will be poor. The second part is more complicated since it includes the volatile nature of traffic speed data. Traffic speeds, particularly at lower sampling frequencies such as 5 or 15 min, can fluctuate fast, so there may be underperformance on border speeds when establishing groups with rigid boundaries. For instance, a model designed for predicting speeds ranging from 60 to 80 will have numerous samples labeled 55 or 85. Consequently, boundaries must be adaptable to accommodate such samples.

Determining the boundaries for this method requires consideration of two factors: the volume of data within each speed group and the data’s volatility. The objective is to achieve a uniform percentage of data across each group, ensuring that all groups possess a comparable volume of data. Additionally, each group must include sufficient data for a deep-learning model. Volatility can be controlled by utilizing overlaps among speed groups. An overlap arises when the average of input data, comprising 30 min of data from six weeks before the labeled speed, which serves as the input matrix for our models, is categorized into a different speed group than the label itself. This sample is considered an overlap between the two-speed groups.

The train and test phases have distinct objectives concerning overlaps. We aim for minimal overlaps in test samples to ensure that models predict speeds specific to that group. However, we require a significant degree of overlap in the training dataset to enhance the robustness of the trained model against the overlaps present in the test samples. Training models to achieve robustness against different labels is important; however, it is essential to recognize that the primary objective of this approach is to minimize variations in traffic characteristics within each model and to isolate these characteristics effectively. Thus, the overlap in a dataset should be limited, as excessive overlap would require the model to learn multiple traffic characteristics. Our analysis indicated that the 10 km/h range includes sufficient overlaps to enhance model robustness while excluding other traffic characteristics. The analyses were conducted using the training dataset to prevent data leakage between the training and testing datasets. Due to the differing priorities in the creation of train and test datasets, we utilized two distinct sets of boundaries for their formation, as illustrated in Figure 8.

*Pattern Organization of Train Samples*:We divided training samples along intersecting boundaries to enhance the robustness of trained models against the aforementioned volatility. The values used in this process were the labels of training samples. For instance, in Figure 8, the input sample with a value of 68 is categorized within two intersecting groups: the light-traffic model, defined by boundaries of 50 and 90, and the medium-traffic Model within the boundaries of 40 and 70. Both models will use this sample during the training phase. The intersecting boundaries enhance the trained model’s capacity to predict values beyond initial boundaries and increase the sample size for model training.*Pattern Organization of Test Samples*:Following the training phase, it is important to assign test samples to the model that fits them. Unlike the training phase, we cannot use labels for this choice. To make this choice, we use the historical data in the input matrix. We calculate the average of all historical data used as input values and determine the corresponding model based on this average. The training process is characterized by independent boundaries that do not intersect. For instance, in Figure 8, the average of the input matrix is computed as 64.2, corresponding solely to the Light Traffic Model. We employ this speed group model for our final prediction.

In our dataset, we established the boundaries illustrated in Figure 9, which shows the percentage of samples within each group. The upper section of the figure illustrates the division of test samples. Each bar indicates a speed range of 10 km/h, with the percentage value reflecting the number of samples within that speed range. The colors of these bars indicate the categories assigned to the respective ranges. Red, yellow, light green, and dark green correspond to heavy traffic, medium traffic, light traffic, and no traffic, respectively. No overlap can be observed in the prediction of test samples. The lower section of the figure illustrates the division of the training samples. Each bar continues to represent a speed range of 10 km/h. The bar colors in this figure represent the same groups as in the upper figure. Numerous overlaps can be observed in this figure. For example, when making predictions, a traffic flow of 50–60 km/h represents a medium traffic state; however, in training, we use data within this range for heavy, medium, and light traffic states.

Table 2 presents the results of speed-wise organization achieved using an LSTM network. The analysis of these results indicates that this method improved MAPE by 0.5 units, representing a 4% enhancement. Figure 10 presents the hourly MAPE results for both the base method and the model trained using Speed-wise Pattern Organization. The graphic demonstrates that our method resulted in a 1.0 MAPE score improvement during morning rush hours, with a gain of 2.8 MAPE score observed in the evening rush hours, representing an 11% increase over the previously recorded MAPE score during that period.

### 4.3. Training Models with Month-Wise Pattern Organization

Studies in the literature primarily use the months preceding the test month to train a model, particularly for long-term predictions. Two prevalent strategies are employed for the pre-processing of training datasets. The first method involves a random division of the dataset into training and testing subsets, whereas the second method utilizes the preceding consecutive months as training data. Despite their widespread application, these approaches fail to account for the variations in traffic characteristics over different months. The former strategy is mainly used due to insufficient data, while the latter is used because of the scarcity of research examining the commonalities between months.

Figure 11 depicts the speed data for five consecutive months, starting in May and ending in September. This graphic elucidates the disparities between these months, particularly between September and August. Although these months are sequential, it is crucial to recognize that they are the most different within this figure. This distinction extends not only to speed but also to weather patterns, with August being one of the driest months in Istanbul and September being typically rainy and windy. This difference extends into June and July, but May appears most comparable due to its predominantly rainy conditions in Istanbul. Furthermore, schools are open in June and September, while many individuals take vacations in July and August.

This disparity is also evident in the MAPE scores obtained from regular speed prediction training. Figure 12 illustrates the MAPE scores of an LSTM model trained to forecast one month based on the preceding four months. This graphic illustrates that transitional months between seasons have higher MAPE scores compared to months that closely resemble their preceding months.

Figure 13 illustrates the method of the proposed Month-wise Pattern Organization. We create histograms to determine the similarity between months. This approach effectively captures the distribution of speed data, serving as a reliable representation of traffic. To enhance the representation of temporal patterns, we generate seven histograms for each month, encompassing daily speed data for every day of the week; therefore, we mitigate the influence of distinct patterns associated with each day. The histogram bins start at the minimum speed limit of 10 in our dataset and extend to the maximum speed limit of 132, increasing by 5 at each interval. All values from 441 segments have been merged into these histograms. For each day of the week, we calculate the average of the corresponding 5-minute readings, totaling 288 values per day (24 × 60/5).

Figure 14 presents the average histograms for April and August. The numbers depicted in this image are the mean values of the seven histograms generated and used for similarity assessment. We present the histograms of these two months that overlapped to highlight the differences between them. April exhibits lower average speed numbers, whereas August demonstrates higher average speed values. August does not exhibit any average values below 20. This does not imply that speed values below 20 were not observed in August, but that, on average, August does not exhibit traffic speed values below 20. We employed the Euclidean Distance Method to quantify the distance. When looking for similar months, we compared the 12 months preceding each one. For instance, March 2018 was analyzed in relation to the preceding months, extending back to March 2017.

Figure 15 illustrates a heat map depicting the distance between different months. This heat map corroborates our hypothesis that specific months differ from their preceding months. Specifically, September is found to be more similar to May and April than to the preceding month, August. Table 3 presents the six most similar months for each month of the year. The suggested Month-wise Pattern Organization led to a 0.1 unit increase in MAPE compared to a baseline model that did not have pattern organization, as shown in Table 2.

## 5. Experimental Results

This section assesses the efficacy of three proposed pattern organization approaches and the hybrid approach combining speed-wise pattern organization with month-wise pattern organization methods. The examination involves comparing our methods with widely-used deep learning models against their baseline counterparts that do not include any pattern organization techniques.

### 5.1. Experimental Setup

To prove the effectiveness of our pattern organization schemes for the long-term success of traffic flow prediction, we exploit four well-known deep learning models, CNN, LSTM, Bi-Directional LSTM, and GRU.

Table 4 gives a summary of the hyperparameters employed in our experiments. The second column of this table presents the range of values examined using the grid-search technique to identify the optimal values specified in the corresponding columns for each model type. Our CNN model has two convolutional layers with ReLU (rectified linear unit) activation, both of which are batch-normalized and employ the dropout technique. Subsequent to the convolutional layers, an average pooling layer and a batch-normalized dense layer are employed, along with the application of the dropout technique. Each convolutional layer contains 128 kernels, while the dense layer has 128 hidden nodes. There is a single recurrent neural network layer in LSTM, Bi-Directional LSTM, and GRU. This is followed by a dense layer and an output layer. The LSTM layer comprises 256 units, whereas the bi-directional and GRU layers consist of 512 units each. All of these fundamental model types employ dropout and kernel regularization. We perform batch normalization after each dense layer.

The models in this research were trained on a server equipped with an A5000 GPU (Graphics Processing Unit). On average, training each model for 111 road segments required roughly one day. This indicates that each part underwent training for roughly 13 min. We used early stopping, which allowed each model to undergo training for a varying number of epochs; on average, each model underwent training for 800 to 1500 epochs with a batch size of 2048. Conversely, evaluating a sample requires less than a second, so the system’s performance is satisfactory for practical application.

### 5.2. Experimental Results for Proposed Pattern Organization Methods

Figure 16 illustrates the MAPE results of the proposed pattern organizations implemented on popular deep learning models, LSTM, CNN, Bi-directional LSTM, and GRU. The outcomes of our proposed methods are compared with those of a baseline model that does not utilize pattern organization prior to training. Each proposed approach is examined independently, and additionally, a test was performed to evaluate the compatibility of Month-wise Pattern Organization with Speed-wise Pattern Organization when utilized simultaneously.

The analysis of hourly pattern organization in Figure 17 shows that the LSTM and bi-directional models perform better than the base model at all times. CNN, on the other hand, only performs better in the evening and worsens during the other hours, while GRU consistently performs worse at all times.

In terms of Speed-wise Pattern Organization, Figure 16 illustrates that every model type exhibits improvement, albeit the extent of this improvement varies among different model types. LSTM exhibits a greater enhancement than the other model types, achieving a percentage increase of 4, whereas CNN demonstrates the least improvement, with a percentage increase of 2.5. The CNN’s incapacity to learn from smaller datasets may account for this outcome. Although GRU has only enhanced by 3.2%, this improvement occurred at elevated MAPE values, indicating that it has acquired more substantial information.

Figure 18 demonstrates that while the MAPE improvement appears minor on average, it is substantially greater in high-traffic points, demonstrating the effectiveness of our strategy. All trained model types exhibit similar improvements, despite variations in their magnitude. This suggests that our approach simplifies the learning of elements that the various base models struggle to identify. Therefore, pattern organization is suitable for a wide range of models, as it facilitates the understanding of various traffic features.

Figure 16 shows that for Month-wise Pattern Organization, the average improvement over the base model is less than 1%. Although the calculated average is small, Table 5 demonstrates that monthly improvement varies based on the dissimilarity of the tested month to the preceding months. Forecasting distinct months such as September or March yielded a reduction in the MAPE score of approximately 0.3, reflecting a 1.7% enhancement in performance. Another point to consider is that although some months may not show improvement, they do not show a decline.

In the preceding section, we discussed the combination of multiple pattern-organizing techniques. We tested this idea by combining speed-wise and Month-wise Pattern Organization. The two strategies are not mutually exclusive, as one utilizes speed values from the training dataset, while the other modifies the months chosen as the training dataset. Figure 16 illustrates that optimal outcomes are achieved using this combination. GRU attains the highest overall success with a MAPE score of 13. The application of both Speed-wise Pattern Organization and Month-wise Pattern Organization approaches yields a 0.51 improvement in the MAPE score relative to a model devoid of pattern organization techniques. This contribution represents an enhancement of 3.77%; however, the most significant improvement is attributed to the LSTM model. The application of the LSTM model results in a 4.8% reduction in the MAPE score and an improvement of 0.66 points. The CNN and bi-directional models exhibit improvements of 3.2% and 3.7%, respectively. The distinction between this combination and a Speed-wise Pattern Organization parallels the difference between the base model and training employing a Month-wise Pattern Organization. Both of these methods help the model learn different patterns, so mixing them does not take away from the benefits of either one alone.

## 6. Discussion

The goal of this study, its findings, and the suggested method is to make it easier to learn about traffic patterns that occur as a result of external influences. The finite capacity for information collection and processing may result in certain patterns remaining undiscovered by deep learning algorithms. This paper’s methodology involves structuring traffic speed data to facilitate the learning process and enhance the capacity of deep learning algorithms to identify difficult-to-detect patterns. Contemporary research addressing the issue of long-term traffic speed prediction may fail to cover these patterns, as it emphasizes spatial–temporal correlations in traffic data while neglecting underlying patterns that lack an apparent rationale. Our research focuses on this issue in order to close this gap in the literature.

The presented strategies, while efficient in enhancing model performance, do not yield uniform results. The Hour-wise Pattern Organization method does not improve system performance during regular hours, whereas the Speed-wise Pattern Organization demonstrates superior overall performance as it is independent of the time elements. The Month-wise Pattern Organization demonstrates its effectiveness during transition months but fails to improve model accuracy in regular months and in fact worsens the model performance in some months. This tendency to improve more abnormal data while struggling with regular patterns is also seen in all the proposed approaches, although in the hour-wise organization, some hours such as 10 am and 7 pm are the most affected. The proposed methods concentrate on unpredictable chaotic moments. However, they are unable to substantially improve the model’s performance if the predicted road follows a regular pattern, despite their excellent performance on these samples. This constraint suggests that our methods work best on routes that experience sporadic events within the designated timeframe, or when applied to restricted data that does not fully represent the patterns in the dataset.

The proposed strategy for organizing monthly patterns is to identify related months that have similar characteristics. This concept, while accurate in its premise, is too inflexible to adequately capture the essence of changing traffic characteristics. Taking time periods in months fails to account for the shifts that are intended to be separated. For example, the first few weeks of September or March are not the same as the last few weeks and have distinct traffic characteristics. Future research could build on this hypothesis by investigating dynamic seasonal periods that are not limited to months or weeks.

A further limitation of Month-wise Pattern Organization is the representation of months we have used. Our work concentrates on anomalous instances; therefore, we opted to depict each month in a manner that emphasizes these irregularities. The inability to demonstrate consistent improvement on a monthly basis can be addressed by enhancing the quality of representation, which may facilitate the identification of subtle variations in traffic characteristics that can be measured in minutes. Such representations would most likely focus on external data such as weather, events, accident profiles, and so on, as opposed to the suggested methodology, which attempts to capture these underlying factors when they cause significant fluctuations.

In the Hour-wise Pattern Organization method, enhancing the performance of regular hours is more challenging than in the Month-wise Pattern Organization. Enhanced time segmentation may potentially augment the efficacy of the methods; for instance, dynamically adjusting the time frame based on the day of the week or month could enable the creation of groups that more effectively isolate traffic characteristics.

The Speed-wise Pattern Organization performed well when compared to regular time periods; however, samples with different labels and input averages present a significant challenge to this strategy. While we have mitigated the effects of these samples, they remain the primary bottleneck for this technique. Further research can examine the effects of employing adjacent models for prediction, for example, using light and heavy traffic models when the input average corresponds to the medium model. With this approach, volatile samples may be predicted more accurately as the correct model type is included in the process.

A further constraint of our study is the dependence on data from a singular metropolitan area, thus constraining the generalization of our findings to other places. The Hour-wise Pattern Organization periods, Speed-wise Pattern Organization boundaries, and Month-wise Pattern Organization similarities established are optimal for Istanbul; however, they may differ for other cities or in future years in Istanbul due to evolving traffic characteristics.

The variation and imbalanced nature of traffic speed datasets can lead to biased and inaccurate models, which may result in erroneous decision-making regarding urban management. Our methodologies are currently able to provide effective support for short-term predictions. However, the incorporation of recent vehicular data has the potential to enhance short-term forecasting [66]. Addressing performance issues and improving forecasting accuracy could enhance investment policies and resource management, potentially guiding decision-makers toward more proactive decision-making strategies.

## 7. Conclusions

In this paper, we evaluated our pattern organization strategies using four popular model types and discovered that our techniques significantly improved predictive performance relative to regular training. Our Hour-wise Pattern Organization technique, which focuses on traffic characteristics based on the time of day, improved forecast performance by 1.8%. Speed-wise Pattern Organization, concentrating on speed values directly, improved prediction accuracy by 4%. Meanwhile, Month-wise Pattern Organization, which prioritizes traffic characteristics based on months, improved prediction performance by 1% when used with an LSTM model. We show similar improvements across several model types, confirming the compatibility of our three techniques with other potential models. Furthermore, the integration of speed-wise and month-wise methodologies resulted in an increased improvement percentage of 4.8%, demonstrating the adaptability of our approaches when used in conjunction. Upon analyzing our results on an hourly basis rather than as a general average, we find that the improvements in predictive performance are greater at lower speed levels and indicate that the effectiveness of our methods is beyond what the average improvement suggests. We observed an improvement around 11% with our proposed methods during peak hours. This analysis demonstrates the effectiveness of our methods, as peak hours are critical for traffic forecasts.

In the future, we intend to demonstrate the effectiveness of our methods not only with more intricate model structures such as GCN but also by incorporating external factors such as weather and accidents as inputs. We aim to conduct experiments on various datasets to establish more universal parameters applicable to different cities. Although this research focuses on long-term predictions, our methods are theoretically applicable to short-term predictions as well. One of our future objectives is to demonstrate the effectiveness of our methods for short-term traffic predictions while incorporating additional external data such as traffic lights and bus stops. These parameters, while not beneficial for long-term predictions, are crucial for short-term forecasts and can enhance the proposed algorithms. We generate subsets that contain critical information to train distinct models. However, we can also utilize these subsets in ensemble learning, as our algorithms effectively split the dataset to address the diversity concern of ensemble learning. We can also create subsets with Speed-wise Pattern Organization to show the speed characteristics of a road segment or Hour-wise Pattern Organization to identify similar road segments. This type of analysis could assist decision-makers in urban management during their city planning process.

## Figures and Tables

**Figure 1 sensors-25-01225-f001:**
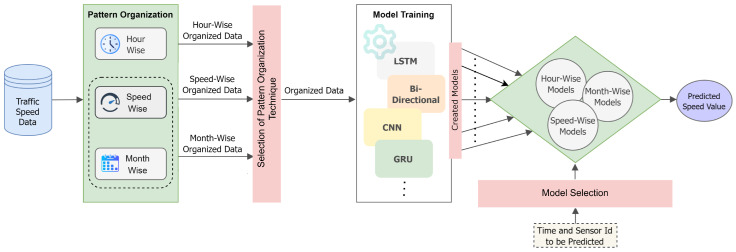
An overview of the general flow of the proposed methodology.

**Figure 2 sensors-25-01225-f002:**
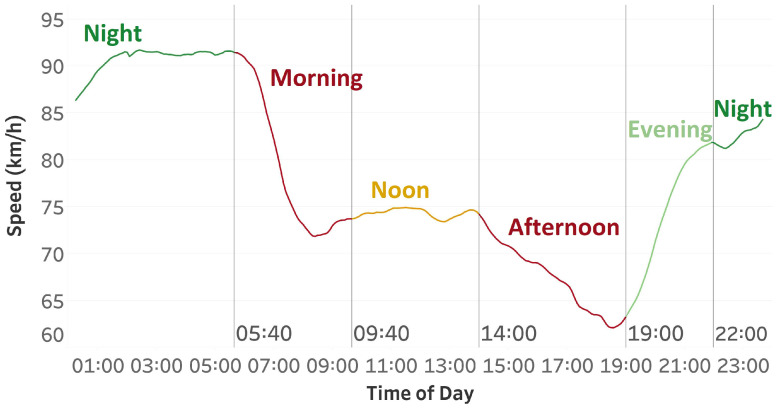
Average Speed of 441 Segments separated to 5 different groups based on time of day.

**Figure 3 sensors-25-01225-f003:**
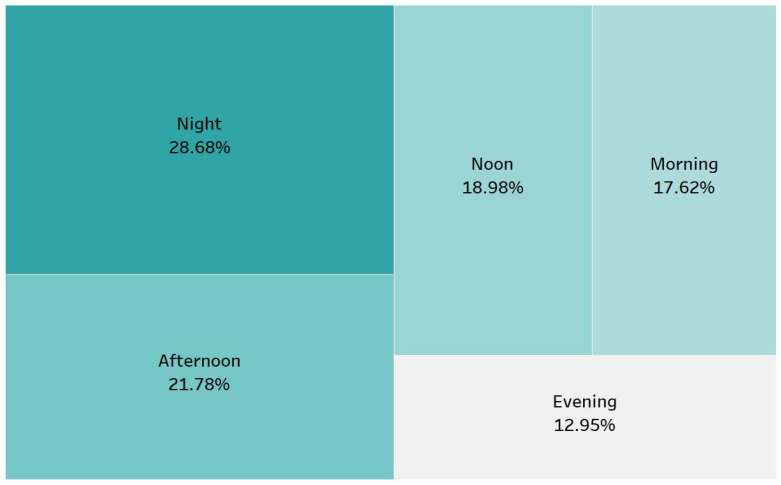
Percentage of data belonging to each time group.

**Figure 4 sensors-25-01225-f004:**
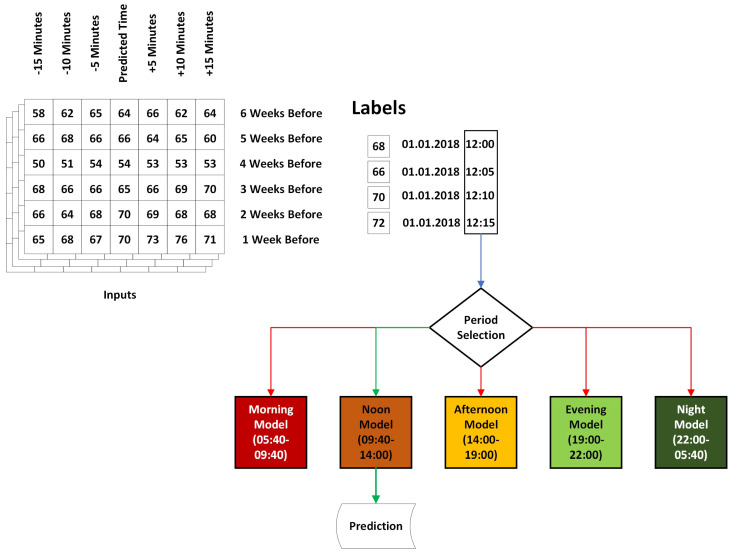
An example of how model creation using Hour-wise Pattern Organization works.

**Figure 5 sensors-25-01225-f005:**
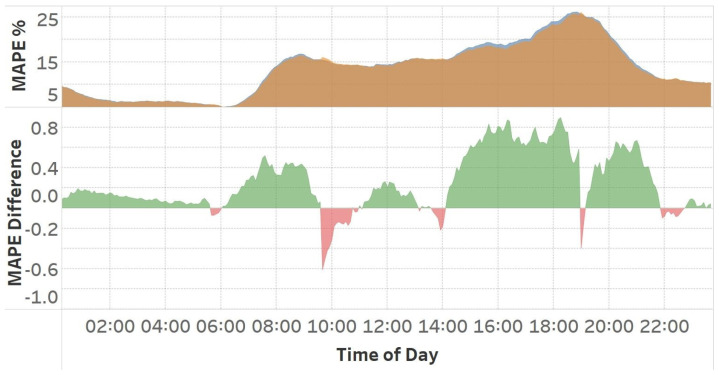
The upper graphics present the MAPE scores of the LSTM model with and without hour-wise pattern organization. The lower part displays the MAPE difference between the two models. Green segments correspond to time regions where the applied method performs better, while red segments belong to regions where the base LSTM model is superior.

**Figure 6 sensors-25-01225-f006:**
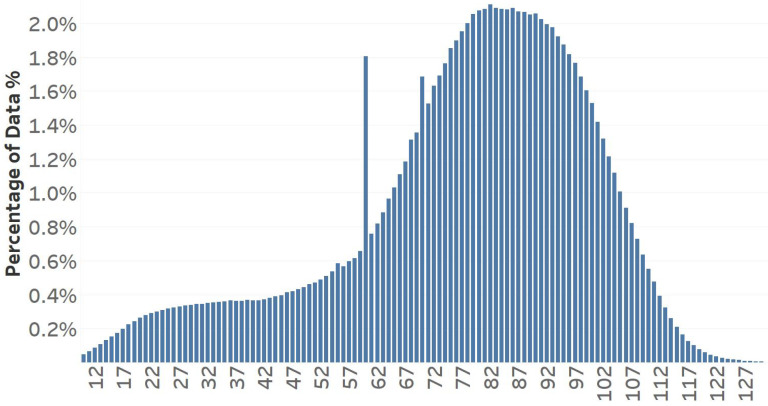
Distribution of speed values within the dataset.

**Figure 7 sensors-25-01225-f007:**
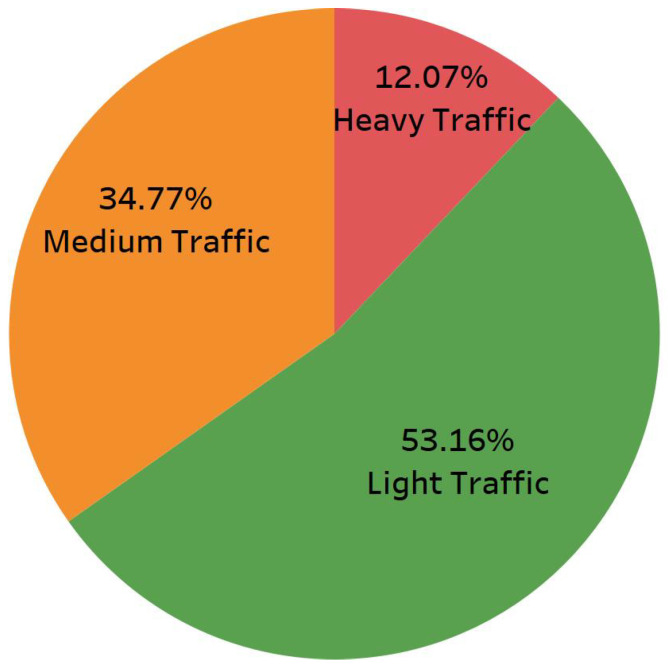
Percentage of data regarding traffic density.

**Figure 8 sensors-25-01225-f008:**
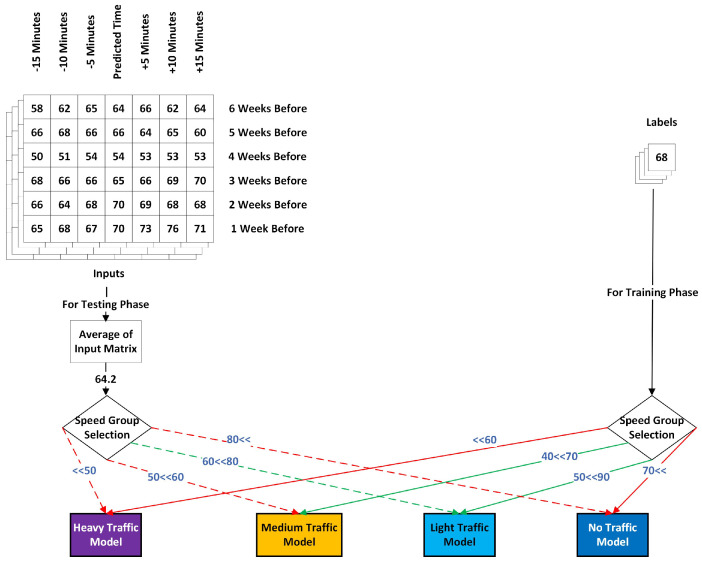
An example of how model creation using Speed-wise Pattern Organization method works.

**Figure 9 sensors-25-01225-f009:**
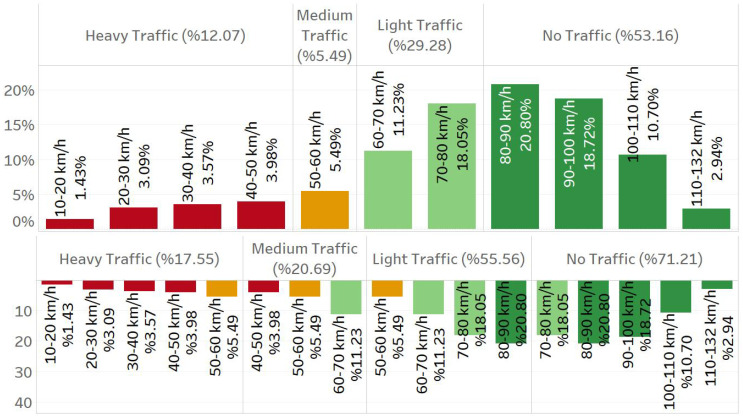
Grouping speed values with the help of speed-wise pattern organization. The upper side of the figure shows the arrangement of the test samples, while the lower side shows the arrangement of the training samples.

**Figure 10 sensors-25-01225-f010:**
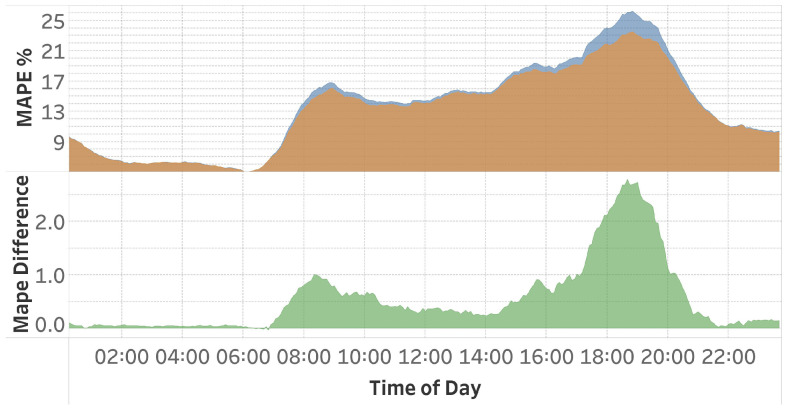
Comparison of the base LSTM model with the model applying Speed-wise Pattern Organization. The upper figure presents the MAPE values of two methods, whereas the lower figure illustrates the MAPE difference between them.

**Figure 11 sensors-25-01225-f011:**
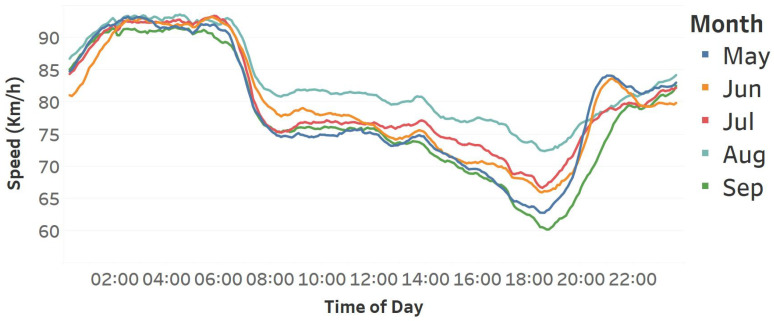
Hourly average speed data of 5 consecutive months belonging to all 441 road segments in Istanbul.

**Figure 12 sensors-25-01225-f012:**
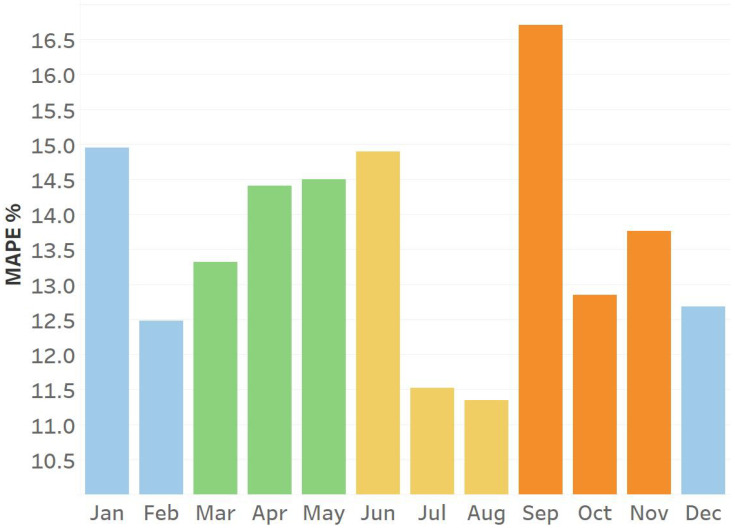
The average monthly MAPE values were obtained from a base LSTM model that was trained without any pattern organization scheme. Each color represents a different season.

**Figure 13 sensors-25-01225-f013:**
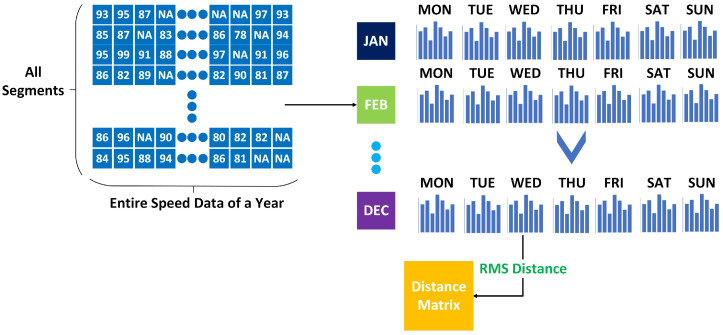
An illustration of the model development process with Month-wise Pattern Organization.

**Figure 14 sensors-25-01225-f014:**
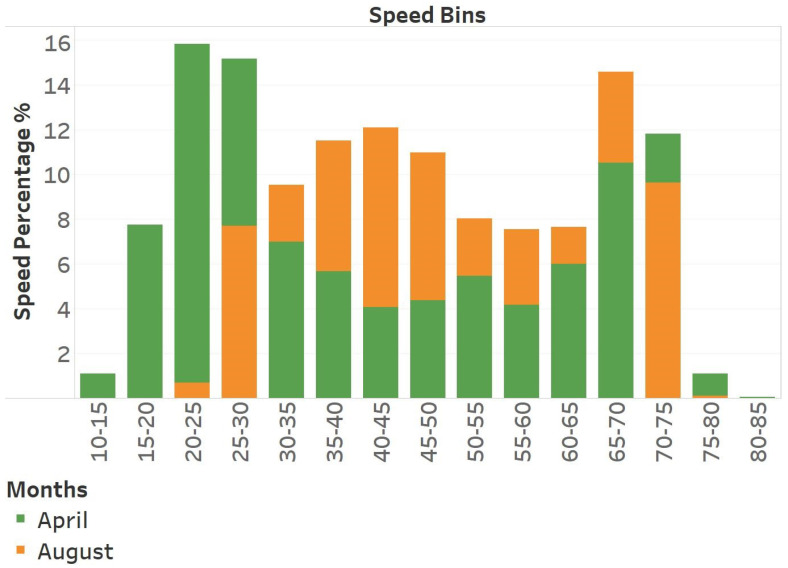
Speed histograms of April and August. We present the speed values of the respective months in bins.

**Figure 15 sensors-25-01225-f015:**
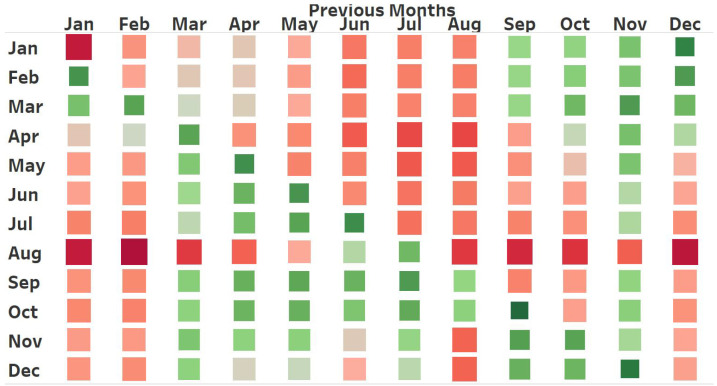
This is a figure illustrating the comparison of month similarities obtained from the proposed methodology. In this heat map, a row depicts the similarity between the month it represents and the twelve previous months shown in the columns. The column designated February indicates the resemblance of the current year’s January and February to February of the preceding year. However, March of this year has similarities to February of this year. The color gradient in this figure progresses from dark red, representing greater distance values, to bright green, denoting lesser distance values.

**Figure 16 sensors-25-01225-f016:**
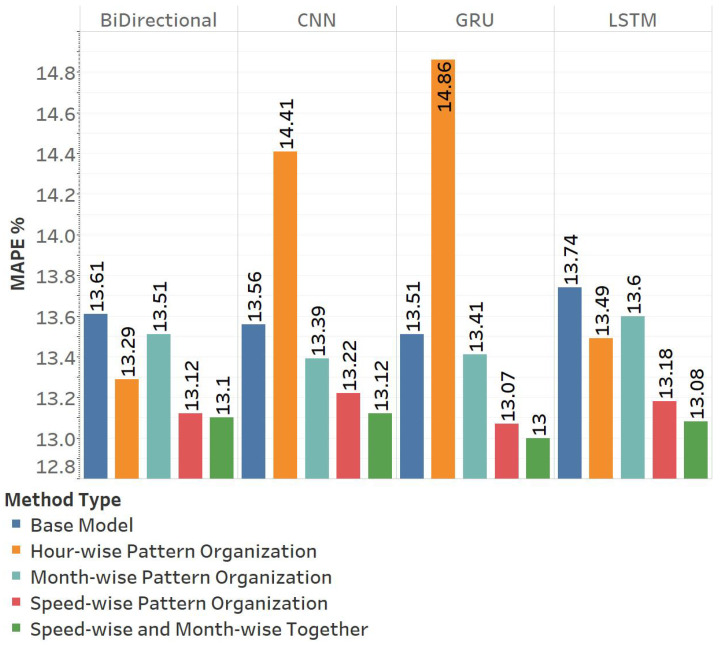
MAPE ratings obtained from four base models. The base models exhibit the MAPE scores obtained without the application of additional methods. The Hour-wise Pattern Organization presents the outcomes obtained using the methodology outlined in Section 3-A. Speed-wise Pattern Organization presents the outcomes obtained using the methodology outlined in Section 3-B. The Month-wise Pattern Organization presents the outcomes obtained using the methodology outlined in Section 3-C. The Multiple Method presents the outcomes obtained using the technique outlined in Section 4-E.

**Figure 17 sensors-25-01225-f017:**
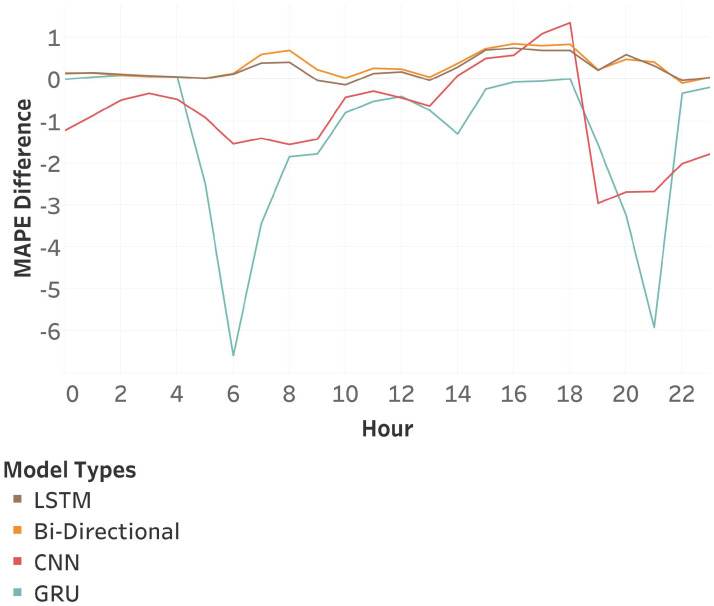
Hourly MAPE Difference between models trained with no additional process and models trained with the hour-wise pattern organization method.

**Figure 18 sensors-25-01225-f018:**
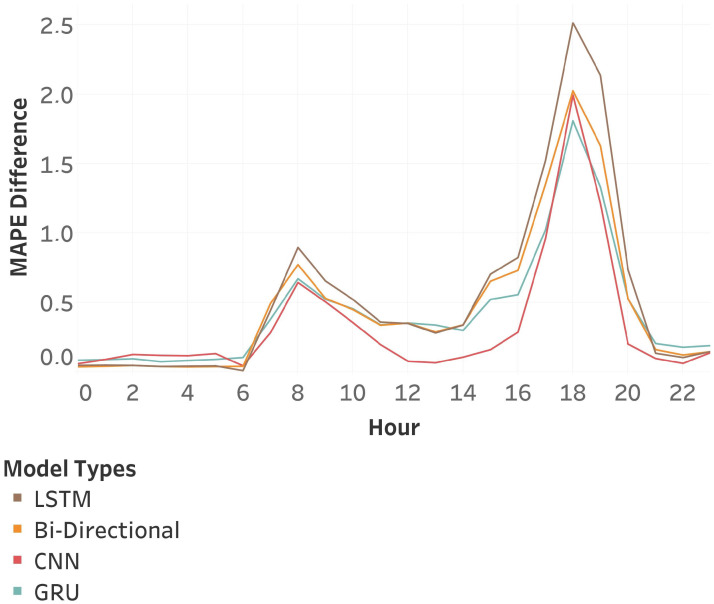
Hourly MAPE Difference between models trained with no additional process and models trained using the Speed-wise Pattern Organization method.

**Table 1 sensors-25-01225-t001:** Dataset and Input Specifications.

Information Name	Value
Data Size for Each Segment	365 Days
Year	2018
Number of Segments	111
Segment Length	1 km
Interval	5 min
Input Shape	7 × 6 (15 min Before and After × 6 Weeks Before Prediction)
Prediction Horizon	1 Week

**Table 2 sensors-25-01225-t002:** We analyze the effects of three proposed pattern organizing methods on the base LSTM model. The average MAPE scores for 111 segments are provided for the proposed preprocessing methods.

	MAPE	Difference with Base Model
**No Pattern Organization**	13.74	
**Hour-wise Pattern** **Organization**	13.49	0.25
**Speed-wise Pattern** **Organization**	13.18	0.56
**Month-wise Pattern** **Organization**	13.6	0.14

**Table 3 sensors-25-01225-t003:** Month similarity information calculated by the proposed method regarding the traffic characteristics of corresponding months.

	Most Similar	2nd Most Similar	3rd Most Similar	4th Most Similar	5th Most Similar	6th Most Similar
**Jan**	Dec	Nov	Oct	Sep	Apr	Mar
**Feb**	Jan	Dec	Nov	Oct	Sep	Mar
**Mar**	Nov	Feb	Dec	Oct	Jan	Sep
**Apr**	Mar	Nov	Dec	Oct	Feb	Jan
**May**	Apr	Nov	Mar	Oct	Dec	Jan
**Jun**	May	Apr	Mar	Nov	Dec	Jan
**Jul**	Jun	May	Apr	Nov	Mar	Oct
**Aug**	Jul	Jun	May	Apr	Nov	Mar
**Sep**	Jul	May	Apr	Jun	Mar	Nov
**Oct**	Sep	Jul	May	Apr	Jun	Nov
**Nov**	Sep	Oct	Mar	May	Apr	Jul
**Dec**	Nov	Sep	Oct	Mar	Jul	May

**Table 4 sensors-25-01225-t004:** Hyper-parameters tested and used in our experiments.

Hyper- Parameters	Tested Range	LSTM	CNN	Bi- Directional	GRU
**Layer Count**	1–3	1 RNN	2 Conv	1 RNN	1 RNN
**Layer Hidden** **Nodes**	32–1024	256	128	512	512
**Dense Layer** **Count**	0–3	1	1	1	1
**Dense Layer** **Hidden Nodes**	32–1024	128	128	128	256
**Dropout** **Percentage**	0–50	40	40	40	40
**Learning Rate**	0.0001–0.1	0.001	0.001	0.001	0.001

**Table 5 sensors-25-01225-t005:** The performance of LSTM, bi-directional LSTM (BD), GRU, and CNN models is affected by month-wise pattern organization (MPO).

	LSTM Base	LSTM MPO	BD Base	BD MPO	CNN Base	CNN MPO	GRU Base	GRU MPO
**Jan**	15.57	15.52	15.41	15.47	15.33	15.29	15.39	15.35
**Feb**	12.82	12.78	12.68	12.68	12.69	12.68	12.78	12.74
**Mar**	13.6	13.42	13.5	13.32	13.43	13.34	13.48	13.3
**Apr**	14.61	14.45	14.49	14.33	15.32	14.3	14.63	14.42
**May**	14.63	14.68	14.45	14.57	14.41	14.61	14.56	14.56
**Jun**	15	14.82	14.76	14.63	14.67	14.6	14.92	14.68
**Jul**	11.68	11.67	11.53	11.58	11.63	11.67	11.67	11.7
**Aug**	11.48	11.41	11.32	11.3	11.36	11.36	11.5	11.49
**Sep**	16.81	16.51	17.06	16.72	15.63	15.39	15.55	15.38
**Oct**	12.91	12.58	12.64	12.44	12.85	12.46	12.58	12.47
**Nov**	13.85	13.61	13.65	13.44	13.48	13.29	13.39	13.27
**Dec**	12.72	12.55	12.56	12.45	12.74	12.56	12.57	12.47

## Data Availability

Data is contained within the article.

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
