# Peer review of "Smart Organization of Imbalanced Traffic Datasets for Long-Term Traffic Forecasting"

_sensors, 2025, doi:10.3390/s25041225_

Round 1
Reviewer 1 Report
Comments and Suggestions for Authors
This paper delved into similarities of traffic speed data: smart organization of traffic patterns to mitigate the imbalance problem in-term traffic forecasting. The paper is interesting and of great importance around long-term traffic forecasting. However, there are some issues that need to be addressed.
1. The title is long. Please, consider shortening.
2. Authors should consider adding their full address including affiliation, email, complete address.
3. Authors stated in their abstract that “we propose three different methods” and “we used four different types of models” Please, specify these methods and models.
4. Acronyms should be defined the first time they appear. What is “MAPE”. Checked throughout the manuscript, all acronyms are defined the first time they appear.
5. Please, consider providing the flowchart of your study.
6. Please, consider providing the managerial implications of your study.
7. What are the limitations of your study? Provide more key points for future studies.
8. Please, check and correct grammatical errors and typos throughout the manuscript.
Comments on the Quality of English Language
Consider improving the English writing through the manuscript
Author Response
Reviewer#1, Concern #1:
This paper delved into similarities of traffic speed data: smart organization of traffic patterns to mitigate the imbalance problem in-term traffic forecasting. The paper is interesting and of great importance around long-term traffic forecasting
Author response: We thank the reviewer for his/her encouraging comments.
Reviewer#1 Concern #2:
The title is long. Please, consider shortening.
Author response: We thank the reviewer for raising this concern. We agree with the reviewer and thus, we updated the title.
Author action: We updated the title as “Smart Organization of Imbalanced Traffic Datasets for Long-Term Traffic Forecasting” instead of “Delving into Similarities of Traffic Speed Data: Smart Organization of Traffic Patterns to Mitigate the Imbalance Problem in Long-Term Traffic Forecasting”.
Reviewer#1, Concern #3:
Authors should consider adding their full address including affiliation, email, complete address.
Author response: We thank the reviewer for pointing out this issue. We added the necessary information.
Author action: We added the full address including affiliation, e-mail, and complete address of the authors.
Reviewer#1, Concern #4:
Authors stated in their abstract that “we propose three different methods” and “we used four different types of models” Please, specify these methods and models.
Author response: We thank the reviewer for pointing this out. We specified these methods and models in Abstract.
Author action: We rewrote the Abstract section as well as specifying three methods and four different models.
"The daily population movement, captured by the time of day, and the weather data, recorded by month, are both considered in this study. Hour-wise Pattern Organization and Month-wise Pattern Organization techniques were devised which organize the speed data using these two factors as a metric with a view to providing a superior representation of data characteristics that are in minority. In addition to these two methods, a Speed-wise Pattern Organization strategy is proposed, which arranges train and test samples by setting boundaries on speed while taking the volatile nature of traffic into consideration. We evaluated these strategies using four popular model types: long short-term memory (LSTM), gated recurrent unit networks (GRU), bi-directional LSTM, and convolutional neural networks (CNN)."
Reviewer#1, Concern #5:
Acronyms should be defined the first time they appear. What is “MAPE”. Checked throughout the manuscript, all acronyms are defined the first time they appear.
Author response: We thank the reviewer for raising this concern. We checked throughout the manuscript and we defined all acronyms.
Author action: All acroynms are defined the first time they appear.
Reviewer#1, Concern #6:
Please, consider providing the flowchart of your study.
Author response: We thank the reviewer for his/her suggestion. We illustrated Figure 1 in order to show the overview of the proposed system.
Author action: We added Figure 1 in order to describe the detail of our study. We also wrote a new paragraph to explain Figure 1.
Reviewer#1, Concern #7:
Please, consider providing the managerial implications of your study.
Author response: We thank the reviewer for his/her suggestion. We added necessary comments both in Discussion part and Conclusion Part.
Author action: We added the last paragraph of the Discussion part and last part of the Conclusion.
Reviewer#1, Concern #8:
What are the limitations of your study? Provide more key points for future studies.
Author response: We thank the reviewer for his/her suggestion.
Author action: We have already defined the limitations of our study in Discussion part. We also added some key points for future work in Conclusion part.
Reviewer#1, Concern #9:
Please, check and correct grammatical errors and typos throughout the manuscript.
Author response: We thank the reviewer for raising this concern. We did our best to fix grammar mistakes and typos. We also revised many parts to improve the quality of the manuscript.
Author action: The writing and language in the paper were extensively reviewed and edits were made. Additionally, the manuscript is checked with a professional tool to fix grammar mistakes and typos.
Reviewer 2 Report
Comments and Suggestions for Authors
The manuscript proposes models to better learn the complex traffic characteristics and also tackling the problem of existing imbalanced dataset data to short timeframe of rush hours. It is a good and interesting work.
The following comments to be addressed:
- The title of the manuscript does not reflect the the problem and objective. Rewrite it please and avoid unnecessary words.
- In abstract the metric MAPE score is used but the reader has no idea about it as it is not defined yet. Replace it with something meaningful.
- The abstract use "organizing traffic patterns" as a solution but it is unclear!! you need to clarify further.
- lack of enough citations in the first paragraph of introduction.
- Include a discussion on analysis of your model when traffic lights are adaptive and can easily change the traffic flow?
- How about when you have live traffic and mobility pattern of vehicles are already available through internet of vehicles. Have a discussion about it. Consider: https://ieeexplore.ieee.org/abstract/document/9445064
- The abstract to focus further on the results.
Author Response
Reviewer#2, Concern # 1:
The manuscript proposes models to better learn the complex traffic characteristics and also tackling the problem of existing imbalanced dataset data to short timeframe of rush hours. It is a good and interesting work.
Author response: We thank the reviewer for his/her encouraging comments.
Reviewer#2, Concern # 2:
The title of the manuscript does not reflect the the problem and objective. Rewrite it please and avoid unnecessary words.
Author response: We thank the reviewer for raising this concern. We rewrote the title.
Author action: We updated the title as “Smart Organization of Imbalanced Traffic Datasets for Long-Term Traffic Forecasting” instead of “Delving into Similarities of Traffic Speed Data: Smart Organization of Traffic Patterns to Mitigate the Imbalance Problem in Long-Term Traffic Forecasting”.
Reviewer#2, Concern # 3:
In abstract the metric MAPE score is used but the reader has no idea about it as it is not defined yet. Replace it with something meaningful.
Author response: We thank the reviewer for his/her suggestion. We defined MAPE and as well as the all acronyms the first time they appear.
Author action: All acroynms including MAPE are defined the first time they appear.
Reviewer#2, Concern # 4:
The abstract use "organizing traffic patterns" as a solution but it is unclear!! you need to clarify further.
Author response: We thank the reviewer for giving us the opportunity to clarify it. We rewrote Abstract in order to define the proposed smart organization mechanism.
Author action: We rewrote Abstract in order to clarify the given expression, i.e., “organizing traffic patterns”.
Reviewer#2, Concern # 6:
lack of enough citations in the first paragraph of introduction.
Author response: We thank the reviewer for raising this concern. We added two references published in 2024.
Author action: We cited the following papers in the first paragraph of introduction.
Do, V.M.; Tran, Q.H.; Le, K.G.; Vuong, X.C.; Vu, V.T. Enhanced Deep Neural Networks for Traffic Speed Forecasting Regarding Sustainable Traffic Management Using Probe Data from Registered Transport Vehicles on Multilane Roads. Sustainability 2024, 16, 2453. https://doi.org/10.3390/su16062453
Cao, C.; Bao, Y.; Shi, Q.; Shen, Q. Dynamic Spatiotemporal Correlation Graph Convolutional Network for Traffic Speed Prediction. Symmetry 2024, 16, 308. https://doi.org/10.3390/sym16030308
Reviewer#2, Concern # 7:
Include a discussion on analysis of your model when traffic lights are adaptive and can easily change the traffic flow?
Author response: We thank the reviewer for his/her suggestion. We discussed it as a future work in Conclusion part.
Author action: We added the last paragraph of the Conclusion.
Reviewer#2, Concern # 8:
How about when you have live traffic and mobility pattern of vehicles are already available through internet of vehicles. Have a discussion about it. Consider:
https://ieeexplore.ieee.org/abstract/document/9445064
Author response: We thank the reviewer for his/her suggestion. We added necessary comments on it in Discussion.
Author action: We added necessary comments and cited the mentioned study in the Discussion part.
- Iranmanesh, F. S. Abkenar, A. Jamalipour and R. Raad, "A Heuristic Distributed Scheme to Detect Falsification of Mobility Patterns in Internet of Vehicles," in IEEE Internet of Things Journal, vol. 9, no. 1, pp. 719-727, 1 Jan.1, 2022, doi: 10.1109/JIOT.2021.3085315.
Reviewer#2, Concern # 9:
The abstract to focus further on the results.
Author response: We thank the reviewer for his/her suggestion. We included numerical results about the study in Abstract.
Author action: We rewrote Abstract and shared more results of the proposed models.
Round 2
Reviewer 1 Report
Comments and Suggestions for Authors
The authors have addressed all my concerns which have significantly improved the manuscript, thereby I recommend the paper to be accepted as is
Author Response
We thank the reviewer for his/her encouraging comments. We would like to express our sincere gratitude.